# Microperimetry-Assessed Functional Alterations and OCT-Changes in Patients after Retinal Detachment Surgery Using Pars Plana Vitrectomy and SF6 Tamponade

**DOI:** 10.3390/diagnostics11071157

**Published:** 2021-06-24

**Authors:** María D. Díaz-Barreda, Isabel Bartolomé-Sesé, Ana Boned-Murillo, Antonio Ferreras, Elvira Orduna-Hospital, Francisco J. Ascaso, Isabel Pinilla

**Affiliations:** 1Department of Ophthalmology, Lozano Blesa University Hospital, 50009 Zaragoza, Spain; mddiaz@salud.aragon.es (M.D.D.-B.); issbartolome@gmail.com (I.B.-S.); anaboned4b@gmail.com (A.B.-M.); jascaso@gmail.com (F.J.A.); 2Aragón Health Research Institute (IIS Aragón), 50009 Zaragoza, Spain; aferreras@salud.aragon.es (A.F.); elvisabi14@hotmail.com (E.O.-H.); 3Department of Surgery, School of Medicine, University of Zaragoza, 50009 Zaragoza, Spain; 4Department of Ophthalmology, Miguel Servet University Hospital, 50009 Zaragoza, Spain; 5Department of Applied Physics, University of Zaragoza, 50009 Zaragoza, Spain

**Keywords:** rhegmatogenous retinal detachment, microperimetry, retinal thickness, optical coherence tomography, swept source OCT

## Abstract

Background: We study the retinal function measured by macular integrity assessment microperimetry (MAIA) and structural changes assessed by scanning swept source optical coherence tomography (SS-OCT) between healthy individuals and patients undergoing pars plana vitrectomy (PPV) after rhegmatogenous retinal detachment (RRD). Methods: Cross-sectional study. Early Treatment Diabetic Retinopathy Study (ETDRS) grids were measured by SS-OCT and compared with the MAIA parameters. Results: Thirty-eight eyes with RRD (19 macula-on and 19 macula-off) were compared with 113 healthy eyes. The retinal sensitivity and average total threshold were reduced in all sectors in the RRD group; macular integrity index was increased. Macular thicknesses in total retina and ganglion cell layer (GCL)++ protocols were higher in the RRD group in nasal outer (NO) and central (C) sectors and only in C sector for GCL+ protocol. Thicknesses were lower in total retina, GCL++ protocols in the temporal outer (TO) sector and in the GCL+ protocol in NO sector. Best-corrected visual acuity (BCVA) correlated moderately with retinal sensitivity in all sectors and in just several sectors with time between the date of surgery and the test. The central nasal (CN) sector thickness and the average total threshold were higher in the macula-on subgroup. Conclusions: RRD and subsequent surgery results in functional and structural changes, especially in individuals with macular detachment.

## 1. Introduction

Rhegmatogenous retinal detachment (RRD) is the most common type of retinal detachment with an incidence between 9.5 and 18.2 cases per 100,000 inhabitants per year [1]. Pars plana vitrectomy (PPV) along the gas tamponade achieves high anatomical success [2]. Despite great advances in RRD surgery with anatomic reattachment close to 100%, functional improvement is not always as expected since it comes with possible visual acuity (VA) loss, contrast sensitivity, color vision alterations or metamorphoses. These changes negatively affect the quality of life due to visual impairment [3,4,5]. RRD recovery behaves differently according to whether or not the macula is involved; the impairment the fovea and the central 10 degrees of the retina could cause a significant detriment to the patient’s quality of life [6]. The importance of macular involvement is not well-known for the end retinal sensitivity result [7]. Presurgical variables, such as previous best-corrected VA (BCVA); time between the onset of symptoms and surgery; extension of the RRD; macular involvement before surgery; vitreomacular traction and distance between the outer limiting membrane and photoreceptors outer segments measured by optical coherence tomography (OCT) may affect the final best-corrected visual acuity (BCVA) and other functional parameters [4,5,8]. Postsurgical variables, such as cystoid macular edema or retinal folds; epiretinal membrane formation; retinal pigment epithelium (RPE) migration; rupture and loss of the photoreceptors; and persistence of subretinal fluid may also affect the final vision [3,9]. Histological modifications after RDD have been studied in animal models and they have proven that in the first 24 h after RRD, RPE changes are already occurring [10]. After 24 h to 72 h, photoreceptor outer segments begin to degenerate and retinal changes progress over time with inner segment degeneration [11], changes in the outer nuclear layer and outer plexiform layer and neuronal remodeling [12,13]. The correct juxtaposition of the RPE and neurosensorial retina can stop the process, although it is not clear whether this can achieve involution and recovery, either partial or complete, of the affected cells. [14]

The introduction of OCT has allowed us to study microstructural changes in the retinal layers [15,16]. 

Tests to assess retinal function have also improved remarkably. Macular function can be studied using either psychophysiological or electrophysiological test. Psychophysical tests include VA, color vision, Amsler grid, photostress and microperimetry among others. They offer different characteristics of macular function including minimum angular separation, color perception, defect in the central field, photoreceptor recovery with respect to bright lights or central retinal sensitivity [17]. In recent years, the macular integrity assessment (MAIA) microperimetry was introduced to clinical practice. MAIA provides several improvements over conventional and kinetic perimetry [18,19,20], such as the possibility of seeing the retina in real-time while performing the test [21], analysis of retinal sensitivity and the ability to analyze fixation with the “eye tracker” system [19,22]. 

The aim of the present study was to evaluate and compare the retinal thickness changes measured by swept source optical coherence tomography (SS-OCT) and the functional parameters obtained with MAIA microperimetry between eyes that underwent RRD surgery depending on the previous macula status and healthy eyes.

## 2. Materials and Methods

The study protocol adhered to the tenets of the Helsinki Declaration. The study complied with Spanish legislation in the field of biomedical research; in the protection of personal data, Organic Law 3/2018 on the Protection of Personal Data; Basic Law 41/2002, regulating patient autonomy and rights; obligations regarding information and clinical documentation; and Law 14/2007 on biomedical research. The study was approved by the Aragon Clinical Research Committee with ID PI19/253.

We conducted a unicentric cross-sectional study at the Ophthalmology Department of the Hospital Clínico Universitario Lozano Blesa, Zaragoza, from October 2018 to October 2019. Patients who had undergone RRD surgery were included in the study. They were asked to attend regular check-ups at the Ophthalmology Department during the follow-up and they agreed to take part in the study by signing the Informed Consent form. A total of 38 patients possessing primary RRD with fewer than 2 weeks between the onset of symptoms and surgery and the absence of other ocular pathologies that could compromise the BCVA with successful surgical reattachment after 23G PPV plus sulfur-hexafluoride (SF_6_) tamponade were included. Nineteen of the subjects were macula On RRD and 19 had macular involvement (macula Off). The time elapsed between surgery and study testing ranged from 15 days to 4 years. All the tests were performed with a complete disappearance of the intraocular SF_6_. The post-surgical BCVA had to be below 1.0 (logMAR scale). Exclusion criteria included proliferative vitreoretinopathy above grade B; ocular pathology affecting central vision (age-related macular degeneration, pathological myopia, macular hole, epiretinal membrane, diabetic retinopathy with clinically significant macular oedema, glaucoma diagnosed with perimetric involvement, papillary atrophy, amblyopia and macular scarring); intraocular pressure (IOP) over 20 mmHg; and the lack of fixation or cooperation during the exam.

Control group included family members or healthcare personnel with no previous history of eye disease. All of them signed the Informed Consent form. 

All participants underwent a comprehensive ophthalmologic evaluation, including date of birth, sex, ophthalmologic history, refractometry, axial length (AL) and spherical equivalent (SE) measured with the ALADDIN KR-1W Series optical biometry system (Topcon Corporation, Tokyo, Japan), BCVA (measured according to the logMAR scale), intraocular pressure with a Goldmann applanation tonometer, macular status (on/off) and checking for the absence of vitreoretinal proliferation of any grade. 

Macular retinal sensitivity was evaluated by MAIA (MAIA, Macular Integrity Assessment system, CenterVue SpA, Padova, Italy) carrying out an Expert Exam following 4-2 strategy. It provides an analysis of 37 retinal sensitivity points corresponding to specific stimulated locations represented by a color scale map. It also reports data on fixation losses (percentage), macular integrity index, average total threshold, fixation stability, bivariate contour ellipse area (BCEA) and BCEA 63% and 95% angles were measured. The macular integrity index corresponds to a numerical value ranging from 0 to 100 indicating whether a patient’s response is normal (close to 0), suspicious or abnormal (as it approaches to 100). Depending on the number of measured fixation points within a circle having radius of 2° (p1) or 4° (p2), fixation stability was considered stable (p1 and p2 > 75%), relatively unstable (p1 < 75% and p2 > 75%) or unstable (p1 and p2 < 75%). BCEA corresponds to the area of 2 ellipses perpendicular to one another containing 63% (minor ellipse) or 95% (major ellipse) of the fixation points, respectively. The angle between them indicates the direction in which the eye movements are oriented during the test. These data are shown in Figure 1.

After dilating the pupil with tropicamide eye drops (Tropicamida^®^, Alcon Cusi, Barcelona, Spain) a fundus examination and macular SS-OCT (Deep Range Imaging (DRI)-Triton *SS-OCT*, (Topcon Corporation, Tokyo, Japan) were performed. A macular 6.0 × 6.0 mm three-dimensional scan was obtained and the automatic segmentation of each retinal layer was performed by the IMAGEnet 6 software Version 1.22.1.14101© 2014 (Topcon Corporation, Tokyo, Japan). The total retinal thickness, the GCL+ (ganglion cell layer (GCL) plus inner plexiform layer (IPL)) thickness and ganglion cell complex (GCC; GCL++ comprising GCL+ plus retinal nerve fiber layer (RNFL)) thickness were measured. 

In order to correlate the data provided by MAIA and SS-OCT, we divided the 37 points provided by microperimetry into sectors similar to those automatically generated by OCT following the Early Treatment Diabetic Retinopathy Study (ETDRS) grid. Assuming that for emmetropic eyes, a degree would be approximately equivalent to a radius of 0.3 mm, 3° to 0.9 mm and 5° to 1.5 mm [23,24]. In this manner, the central point and the 12 points of the first concentric circle would correspond to the central sector (C) of the OCT. This is reflected in Figure 2. 

In the MAIA parameters, we differentiated the central point, referred to as C, and the 12 surrounding points in the 4 quadrants. We referred to these 12 points with the letter C as the central sector, followed by the letter S for the superior sector, T for temporal, I for inferior and N for nasal. In addition, we averaged the 13 points included in the central sectors and we termed the value C global. The thresholds of the radius of 3° (inner (I)) and 5° (outer (O)) of the MAIA would correspond to the inner ring or parafovea of the OCT. We subdivided them into 4 quadrants (S, T, I, N). To obtain a figure that allows us to compare the results of each with the other and we averaged the sensitivity threshold of the points included in each sector. The microperimetry fundoscopic images were exported from the MAIA and incorporated into the SS-OCT for later analysis and comparison.

Elapsed time from the date of surgery to the date when the structural and functional tests were performed was also recorded.

### Statistical Analysis

Statistical analysis was performed using SPSS (SPSS 25, SPSS Inc., IBM Corporation, Somers, NY, USA). Variables are expressed as mean, standard deviation and median and range when the variable was non-parametric. Qualitative variables are expressed as number of cases and as percentages. Normality was analyzed with the Kolmogorov Smirnov’s test. Since most of the parameters were not normal, non-parametric tests were performed. Differences between groups were analyzed with the Mann–Whitney U test for independent samples. The Spearman correlation test was used for the bivariate analysis in the comparison of the structural and functional outcomes. In all analyses, we considered *p* < 0.05 to indicate statistical significance.

## 3. Results

### 3.1. Subsection Comparison of Control Group vs. RRD Group

A total of 151 eyes were analyzed, including 38 RRD eyes and 113 controls. The control group comprised of 55 women (48.7%) and 58 men (51.3%); and 59 right eyes (52.2%) and 54 left eyes (47.8%). The RRD group comprised of 12 women (31.6%) and 26 men (68.4%); and 17 right eyes (44.7%) and 21 left eyes (55.3%). The mean age of the control group was 58.48 ± 9.65 years (33–83) and the mean age of the RRD group was 58.29 ± 9.02 years (22–72; *p* = 0.603). Intraocular pressure was also similar in both groups (control group: 13.59 ± 2.64 and RRD group: 14.42 ± 2.26 mmHg). No differences in sex, laterality, age or intraocular pressure were found between groups. 

BCVA was higher in the control group (0.05 ± 0.08 logMAR) than in the RRD group (0.16 ± 0.18 logMAR; *p* < 0.001). The SE was lower in the RRD group, while AL was lower in the control group. 

The macular integrity index was lower in the control group (*p* < 0.001), while the average total threshold was higher (*p* < 0.001), which indicates a higher probability of finding normal values in terms of sensitivity in the control group. No differences in the fixation stability or BCEA parameters were detected between groups (Table 1). 

Retinal sensitivity determined by MAIA was lower in all sectors in the RDD group as can be seen in Figure 3. It shows the macular sensitivity values provided by the MAIA which are grouped in the sensitivity map according to the color scale. 

In the total retina protocol, an increase in the retinal thickness was observed in the nasal outer (NO) (*p* = 0.001) and the central (C) (*p* = 0.040) sectors in the RRD group compared with the control group while the temporal outer (TO) sector was decreased (*p* = 0.010). Using the GCL+ protocol, the retinal thickness was higher in the C sector (*p* = 0.001) and lower in the NO sector (*p* = 0.048) in the RRD group. In the GCL++ protocol retinal thickness increased in the NO (*p* = 0.001) and the C (*p* = 0.002) sectors and diminished in the TO (*p* = 0.002) in the RRD group (Figure 4).

No correlation was detected between the C global sector evaluated by MAIA and ETDRS C thickness measured with the SS-OCT, regardless of the protocol used. 

The ETDRS sector inferior inner (II) of the GCL+ protocol correlated with the TO (r = 0.197; *p* = 0.015), inferior outer (IO) (r = 0.187; *p* = 0.022), NO (r = 0.199; *p* = 0.014), superior inner (SI) (r = 0.170; *p* = 0.037), temporal inner (TI) (r = 0.162; *p* = 0.047), II (r = 0.191; *p* = 0.019) and nasal inner (NI) (r = 0.191; *p* = 0.019) sectors of the MAIA (Table 2).

Additionally, the ETDRS nasal inner (NI) sector of the GCL+ protocol correlated with the TO sector (r = 0.180; *p* = 0.027) of the MAIA. 

The macular integrity index and average total threshold of MAIA microperimetry were mildly to moderately correlated with age, BCVA and spherical equivalent (Table 3).

BCVA (logMAR) yielded moderate negative correlations with retinal sensitivity determined by MAIA at all sectors. 

### 3.2. Comparison between the Macula-On and Macula-Off Subgroups

The macula-on subgroup included 26.32% women and 73.68% men, while the macula-off subgroup comprised 36.84% women and 63.16% men (*p* = 0.485). Retinal sensitivity measured by MAIA in the central nasal sector and the average total threshold were lower in the RRD subgroup with macula-off (Table 4 and Figure 5). 

Total retinal, GCL+ and GCL++ thicknesses were similar between the subgroups as can be observed in Figure 6.

The number of days that elapsed between the surgery and test date correlated moderately with the logMAR BCVA (r = −0.463; *p* = 0.003) and retinal sensitivity evaluated by MAIA in the superior outer (r = 0.340; *p* = 0.037), temporal inner (r = 0.342; *p* = 0.035), C (r = 0.385; *p* = 0.017), central temporal (r = 0.372; *p* = 0.022) sectors and C global (r = 0.417; *p* = 0.009). 

## 4. Discussion

In our study, macular sensitivity was affected by RRD and related surgical procedures. We studied a consecutive series of RRD treated with 23G PPV, endophotocoagulation and 25% SF6 as tamponade. Retinal sensitivity was reduced in all sectors measured by MAIA in the group of patients undergoing surgery for RRD compared with healthy subjects. Based on anatomic changes measured by structural SS-OCT, the differences were less marked and occurred only in the horizontal areas of the perifoveal ring and in the central area (TO, NO and C). 

The inferior inner sector of the GCL+ protocol, including the GCL–IPL complex, showed mild correlations with all the sectors of the MAIA except the superior outer sector. The nasal inner sector of the GCL+ also correlated with the TO sector of the MAIA. These changes at the inner retinal layers could be explained by the damage caused by the RRD itself, especially in retinal detachments with macular involvement as well as some of the surgical maneuvers: modifications of the intraocular pressure, use of perfluorocarbon liquid, fluid–air exchange or gas injection. Alterations in the outer retina, such as the loss of the photoreceptor outer and inner segments, and photoreceptor cell body or outer plexiform layer changes have been described in animal models or after retinal detachment surgery that may correlate with the visual function. The disconnection between the outer nuclear layer and the RPE generates these first changes [10,11]. Nevertheless, we were unable to find any OCT change at the outer retinal layers depending on the pre-surgical macular status [25], which could be related to the number of studied patients or to the short surgery delay. Only a few studies, however, have focused on inner retina modifications after RRD surgery. Changes in postsynaptic neurons are a common finding in nervous system lesions. Ganglion cell downregulation has been described in an experimental model of retinal detachment and another study observed RNFL thinning over time [26,27]. In our study, we found alterations in the inner retinal layers in patients that underwent RRD surgery and they are predominantly in the GCC or total retinal thickness with fewer effects in the GCL–IPL complex. 

In all studied protocols, GLC+, GCL++ and total retinal thickness were slightly different between both macula-on and macula-off RRD subgroups and between the RRD group and control group; however, the differences were more evident when the RNFL was included in the thickness measurement. We found an increase in retinal thickness in the central and NO areas of RRD group. The activation of the Müller cell after the RRD and the presence of subtle epiretinal membrane or subretinal fluid pockets could justify this increase. It seems that the damage produced after RRD is mainly axonal without a clear loss of ganglion cell bodies. Previous studies described the sequence and mechanisms that results in the loss of the GCL in other diseases such as glaucoma or diabetes [28,29,30].

Montesano et al. [16] reported difficulties in determining whether a particular functional area correlates with ganglion cells located immediately below or spatially distant based on the histologic studies of Drasdo el al. [31]. We did not consider displacement of the ganglion cells from their fields due to the already described variability between subjects. For this reason, we correlated the MAIA and OCT points in their exact locations.

The anatomic and functional changes that occur in the retina after surgery for RRD repair have rarely been evaluated with both OCT and microperimetry. Smith et al. [4] studied a sample of 17 RRD macula-off after surgeries using time-domain OCT (Stratus) and spectral-domain OCT. Of the 17 cases, 11 were functionally evaluated using MP-1 microperimetry and 6 exhibited functional alterations related to different OCT findings. Lai et al. [32] compared their results obtained after performing spectral-domain OCT, autofluorescence and microperimetry (MP-1) in patients undergoing RRD surgery. They found disruption in the outer limiting membrane, photoreceptors and the photoreceptor inner segment/outer segment junction in 43.2% of the cases, while 56.8% presented with blue autofluorescence anomalies related to the OCT findings. The areas of structural, microvascular and functional alterations were similar in both groups and associated with worse BCVA. Scheerlinck et al. [33] studied 40 PPV for RRD by OCT and microperimetry (OptosOCT/confocal scanning laser ophthalmoscopy (SLO)) at 1 and 2 months after surgery. Half of them underwent surgery using SF6 as tamponade and the other half with a worse prognosis underwent surgery with silicone oil tamponade. They found better BCVA and fewer OCT changes in individuals in which SF6 was used. Although there was evidence of GCL and IPL thinning and a possible influence on fovea homeostasis due to their contact with the silicon oil, the patients were not randomized and thus the silicon oil was used in patients with a negative prognosis factor, which may have biased the results. Noda et al. [34] compared pre-surgical and post-surgical (6 months after surgery) MAIA and SS-OCT data in 34 macula-off RRD. They found an improvement in sensitivity after surgery and concluded that postoperative ellipsoid zone continuity was important for good postoperative retinal sensitivity. Borowicz et al. [7] carried out a prospective study with 62 patients after PPV with SF6 tamponade for RRD repairment, 28 with macula-on and 34 with macula-off performing OCT-2000 and MAIA assessment at 3 months and 6 months after surgery. This overcomes two of the major limitations of our study by increasing the sample size and rendering it prospective rather than cross-sectional. They obtained a final BCVA similar to ours for patients with macula-on (0.15 ± 0.20 in their study vs. 0.13 ± 0.1 in our study); however, results were not similar for the case of macula-off eyes. Their BCVA was 0.40 ± 0.28 and ours was 0.19 ± 0.17. Our exclusion criteria included post-surgical vision below 1.0 (logMAR scale) and any factor that could affect central vision because we considered that low VA must affect the MAIA test results and thus rendering the test unreliable. The number of days that elapsed between surgery and the date of the functional and imaging tests was negatively correlated with the BCVA (logMAR), suggesting an improvement in the patient’s visual function. Moreover, the elapsed time showed a positive correlation with different MAIA sectors. This finding suggests that BCVA could improve over time and that functional improvement requires a longer follow-up to be detected than that proposed by Borowicz. They reported no differences related to the elapsed time between the intervention and the testing procedure, except for the P1 value (*p* = 0.02) obtained by MAIA which improved during the follow-up. This could be related to either the structural and functional recovery of the retina or to a learning effect after performing the test more than once. Another reason could be the development derived from the plasticity of the visual system [21,35]. Several studies on patients with different macular pathologies have used devices such as the MP-1 microperimeter (which offers different biofeedback strategies) or even video game stimulation to define a new preferred retinal locus (PRL), which would act as a pseudofovea and optimize residual vision and enhance the fixation. They concluded that flexibility in the ocular motor system exists and could improve the fixation although it does not necessarily have to be accompanied by increased retinal sensitivity [35,36,37]. This could be in favor of an improvement in fixation stability and visual outcomes over time.

We did not find significant differences regarding fixation capacity and stability was not found between the control group and the DRR group nor between the macula-on and macula-off subgroups. Due to the lack of follow-ups, one of the main limitations of our study is that we cannot confirm changes in the fixation data.

In conclusion, RRD and the surgery performed to repair RRD results in functional and structural changes, including alterations in the inner retinal layer and especially in individuals with macular involvement. These changes can be functionally quantified by MAIA and anatomically assessed by SS-OCT. Findings obtained using both techniques correlate with each other. Our findings help to clarify the implications of surgical intervention after RRD and its effect on the patient’s BCVA. Future prospective studies with larger sample sizes would help to strengthen these results. 

## Figures and Tables

**Figure 1 diagnostics-11-01157-f001:**
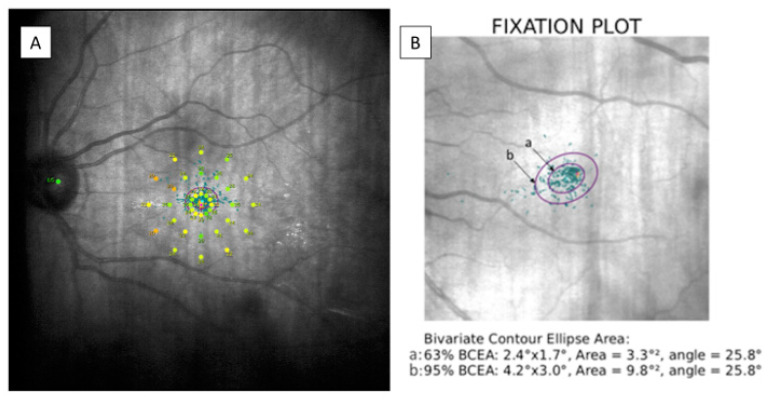
(**A**) Modified image provided by MAIA microperimetry showing the 37 sensitivity points on a color scale map. (**B**) BCEA data corresponding to a patient examined during the study. The BCEA corresponds to the area of two ellipses; “a” containing 63% (minor ellipse) and “b” containing 95% (major ellipse) of the fixation points. MAIA: macular integrity assessment, BCEA: bivariate contour ellipse area.

**Figure 2 diagnostics-11-01157-f002:**
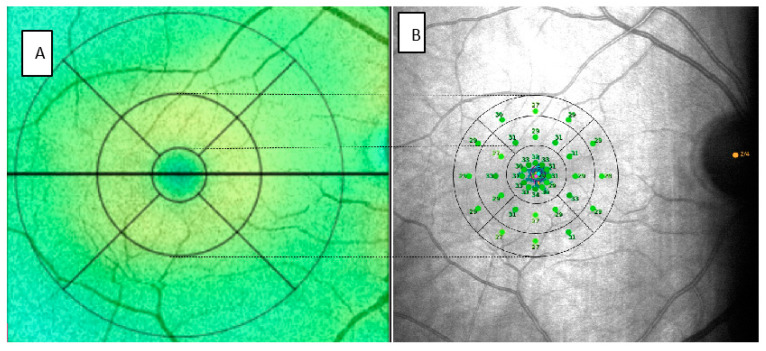
Images provided SS-OCT (**A**) and MAIA (**B**) of the same patient modified to represent the correlation between the two. SS-OCT: swept source optical coherence tomography, MAIA: macular integrity assessment.

**Figure 3 diagnostics-11-01157-f003:**
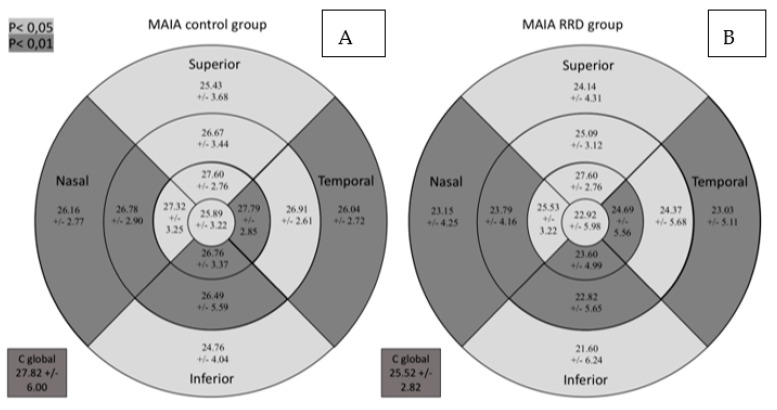
Comparative results between the control group (**A**) and the group of patients with RRD (**B**) for retinal sensitivity measured by MAIA in dB in the study sectors. MAIA: macular integrity assessment; ETDRS: Early Treatment Diabetic Retinopathy Study; RRD: rhegmatogenous retinal detachment; C global: central global. Significant comparisons between the groups (*p* < 0.05) are presented in gray background (*p* < 0.05) or darker gray background (*p* < 0.01).

**Figure 4 diagnostics-11-01157-f004:**
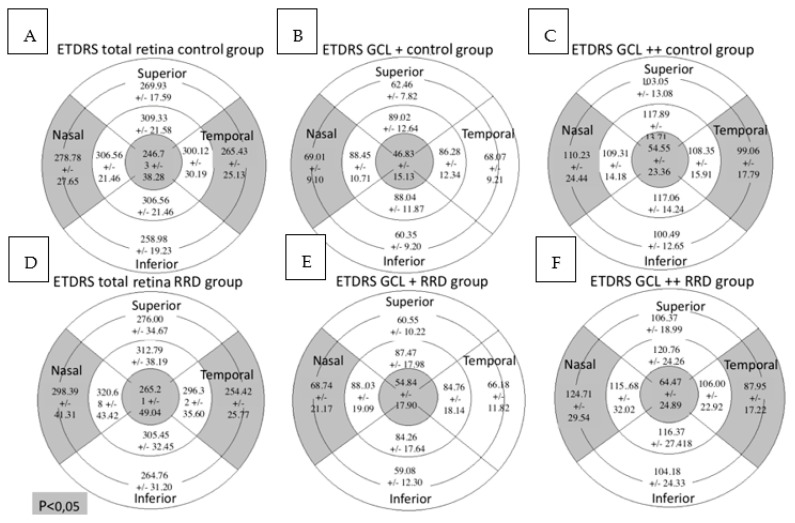
Macular parameters assessed by SS-OCT in both groups. Figures (**A**–**C**) represent control group outcomes while figures (**D**–**F**) represent RRD group outcomes. On top of each figure it is specified the SS-OCT protocol used. Total retinal thickness, GCL+ protocol and GCL++ protocol values are expressed in microns. Mean +/- standard deviation is included in every sector. Significant differences between the groups (*p* < 0.05) are presented in gray background. SS-OCT: swept source optical coherence tomography; MAIA: macular integrity assessment; ETDRS: Early Treatment Diabetic Retinopathy Study; RRD: rhegmatogenous retinal detachment; GCL: ganglion cell layer.

**Figure 5 diagnostics-11-01157-f005:**
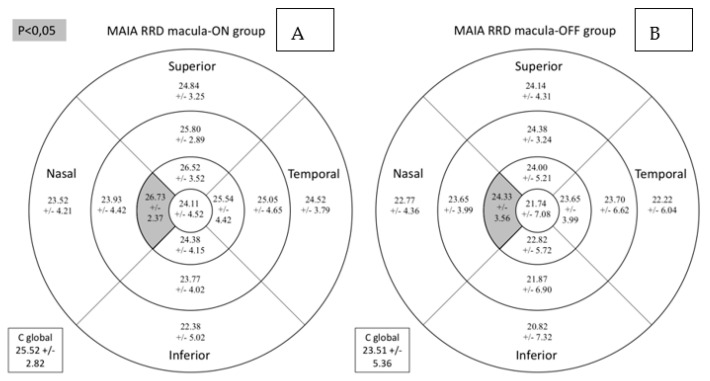
Comparison for retinal sensitivity (dB) assessed by MAIA microperimetry between macula-on (**A**) and macula-off (**B**) subgroups. Mean +/− standard deviation is included in every sector. Significant differences between the groups (*p* < 0.05) are presented in the gray background. MAIA: macular integrity assessment. RRD: rhegmatogenous retinal detachment; C global: central global.

**Figure 6 diagnostics-11-01157-f006:**
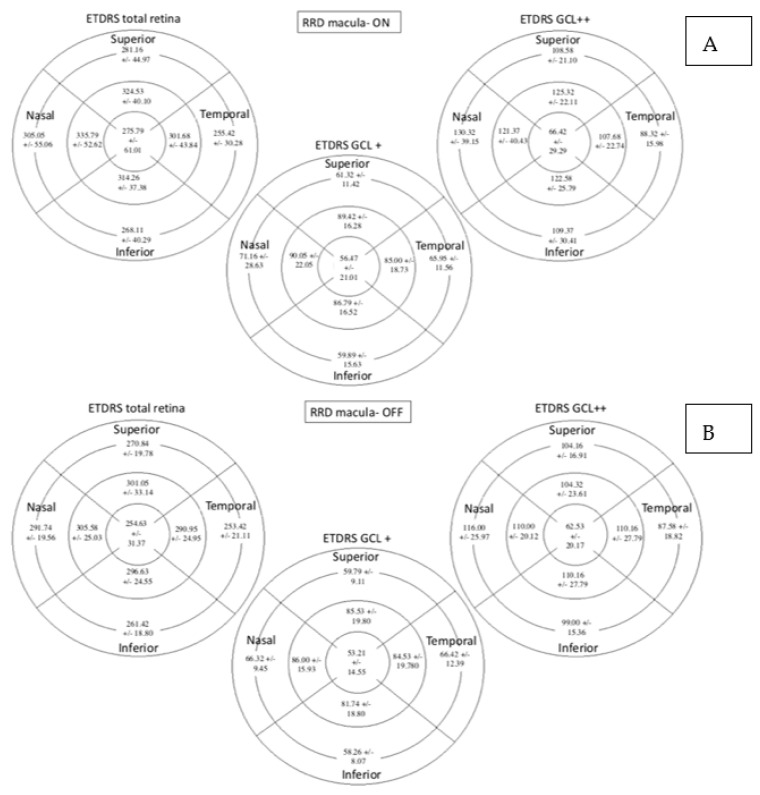
Retinal thickness measured by SS-OCT with the total retina protocol, GCL+ protocol and GCL++ protocol expressed in microns. Figure (**A**) represents macula-On subgroup and Figure (**B**) represents macula-Off. Mean +/− standard deviation is included in every sector. SS-OCT: swept source optical coherence tomography; ETDRS: Early Treatment Diabetic Retinopathy Study; GCL: Ganglion cell layer.

**Table 1 diagnostics-11-01157-t001:** Clinical characteristics of both groups. Significant differences (*p* < 0.05) are highlighted in bold print.

Variable	Control Group	RRD Group	*p*
	Mean +/− SD Median (Range)	Mean +/− SD Median (Range)	
Age (years)	58.48 +/− 9.65 58 (33–83)	58.29 +/− 9.02 59 (33–72)	0.603
BCVA (logMAR)	0.05 +/− 0.08 0.0 (0.0–0.5)	0.16 +/− 0.18 0.1 (0.0–0.6)	**<0.001**
SE (diopters)	4.21 +/− 0.25 0.25 (−12.50–24.87)	−3.78 +/− 4.87 −1.75 (−14.00–1.50)	**<0.001**
AL (mm)	24.38 +/− 2.95 23.63 (21.51–43.89)	26.12 +/− 3.59 25.4 (21.61–44.20)	**<0.001**
IOP (mmHg)	13.59 +/− 2.64 14 (8–20)	14.42 +/− 2.26 15 (9–18)	0.056
Macular Integrity Index	63.08 +/− 29.13 66.9 (2.9–100.00)	80.45 +/− 24.45 90.75 (6.20–100.0)	**<0.001**
Average Total Threshold (dB)	26.44 +/− 2.68 27.1 (18.1–31.3)	24.28 +/− 4.33 25.4 (4.50–29.6)	**<0.001**
Fixation Stability P1 (%)	82.50 +/− 18.35 88 (6–100)	79.34 +/− 26.91 91 (0–100)	0.578
Fixation Stability P2 (%)	94.19 +/− 9.32 97 (27–100)	91.05 +/− 19.38 98 (1–100)	0.347
BCEA 63% area (degrees)	2.80 +/− 3.86 1.6 (0.1–31.2)	4.52 +/− 10.70 1.45 (0.1–62.3)	0.383
BCEA 95% area (degrees)	8.50 +/− 12.16 4.8 (0.2–93.4)	13.52 +/− 32.06 4.3 (0.4–186.7)	0.440
BCEA 63% angle (degrees)	3.98 +/− 60.62 7 (−88.8–89.3)	−6.36 +/− 53.33 −6.9 (−86.7–84.6)	0.319
BCEA 95% angle (degrees)	4.36 +/− 60.63 9 (−88.8–89.3)	−6.36 +/− 53.33 −6.9 (−86.7–84.6)	0.289
Fixation losses (%)	5.94 +/− 13.33 0 (0–75)	3.82 +/− 10.49 0 (0–40)	0.942

RRD: rhegmatogenous retinal detachment; SD: standard deviation; BCVA: best-corrected visual acuity; SE: spherical equivalent; AL: axial length; IOP: intraocular pressure; BCEA: bivariate contour ellipse area.

**Table 2 diagnostics-11-01157-t002:** Spearman correlations between functional data measured by MAIA in different sectors and anatomic data measured by SS-OCT in the 4 sectors the inner ring according to the total retina protocol, GCL+ protocol and GCL++ protocol. Significant correlations (*p* < 0.05) are highlighted in bold print.

		ETDRS SI	ETDRS TI	ETDRS II	ETDRS NI	GCL+ SI	GCL+ TI	GCL+ II	GCL+ NI	GCL++ SI	GCL++ TI	GCL+ II	GCL++ NI
**MP SO**	Correlation coefficient	−0.039	−0.032	0.019	−0.003	0.041	0.038	0.116	0.137	0.012	0.043	0.065	0.046
	*p*	0.634	0.695	0.812	0.966	0.621	0.644	0.157	0.093	0.884	0.602	0.430	0.576
**MP TO**	Correlation coefficient	0.067	0.057	0.120	0.041	0.116	0.064	0.197	0.180	0.045	0.040	0.092	0.072
	*p*	0.417	0.484	0.141	0.619	0.154	0.435	**0.015**	**0.027**	0.584	0.628	0.263	0.378
**MP IO**	Correlation coefficient	0.019	0.052	0.126	−0.010	0.123	0.118	0.187	0.132	0.068	0.106	0.096	0.136
	*p*	0.814	0.527	0.122	0.908	0.134	0.150	**0.022**	0.106	0.409	0.196	0.240	0.096
**MP NO**	Correlation coefficient	−0.016	0.024	0.102	−0.032	0.078	0.066	0.199	0.156	0.008	0.067	0.103	0.068
	*p*	0.844	0.770	0.215	0.694	0.339	0.422	**0.014**	0.055	0.924	0.411	0.210	0.408
**MP SI**	Correlation coefficient	0.063	0.085	0.104	0.031	0.071	0.071	0.170	0.131	0.026	0.085	0.112	0.021
	*p*	0.442	0.299	0.205	0.704	0.387	0.387	**0.037**	0.109	0.753	0.297	0.171	0.796
**MP TI**	Correlation coefficient	0.012	0.034	0.044	−0.059	0.108	0.056	0.162	0.076	0.034	0.043	0.055	−0.053
	*p*	0.879	0.682	0.596	0.472	0.187	0.493	**0.047**	0.356	0.680	0.601	0.504	0.514
**MP II**	Correlation coefficient	0.062	0.096	0.156	−0.003	0.133	0.102	0.191	0.129	0.064	0.082	0.097	0.044
	*p*	0.447	0.242	0.055	0.972	0.104	0.214	**0.019**	0.115	0.437	0.320	0.237	0.588
**MP NI**	Correlation coefficient	0.008	0.050	0.078	−0.028	0.088	0.086	0.191	0.157	0.031	0.078	0.092	0.050
	*p*	0.925	0.538	0.338	0.733	0.280	0.291	0.019	0.055	0.708	0.342	0.263	0.545

ETDRS: Early Treatment Diabetic Retinopathy Study; GCL: ganglion cell layer; MP: microperimetry; SO: superior outer; TO: temporal outer; IO: inferior outer; NO: nasal outer; SI: superior inner; TI: temporal inner; II: inferior inner; NI: nasal inner.

**Table 3 diagnostics-11-01157-t003:** Spearman correlations between functional data measured by MAIA and the clinical variables. Significant correlations (*p* < 0.05) are highlighted in bold print.

		Age	BCVA (LogMAR)	SE	AL	IOP
MP SO	Correlation coefficient	−0.191	−0.370	0.136	0.007	0.092
	*p*	**0.019**	**<0.001**	0.095	0.930	0.262
MP TO	Correlation coefficient	−0.145	−0.416	0.228	−0.121	−0.132
	*p*	0.075	**<0.001**	**0.005**	0.140	0.106
MP IO	Correlation coefficient	−0.122	−0.323	0.206	−0.140	0.009
	*p*	0.137	**<0.001**	**0.011**	0.086	0.912
MP NO	Correlation coefficient	−0.129	−0.382	0.213	−0.120	0.053
	*p*	0.115	**<0.001**	**0.009**	0.143	0.519
MP SI	Correlation coefficient	−0.197	−0.385	0.159	−0.113	−0.004
	*p*	**0.015**	**<0.001**	0.051	0.166	0.964
MP TI	Correlation coefficient	−0.179	−0.425	0.238	−0.167	−0.061
	*p*	**0.028**	**<0.001**	**0.003**	**0.040**	0.460
MP II	Correlation coefficient	−0.142	−0.433	0.254	−0.211	−0.024
	*p*	0.081	**<0.001**	**0.002**	**0.009**	0.773
MP NI	Correlation coefficient	−0.142	−0.391	0.241	−0.159	0.021
	*p*	0.081	**<0.001**	**0.003**	0.052	0.798
MP C	Correlation coefficient	−0.192	−0.356	0.049	−0.043	0.012
	*p*	0.018	**<0.001**	0.550	0.600	0.885
MP CS	Correlation coefficient	−0.128	−0.386	0.220	−0.191	0.000
	*p*	**0.118**	**<0.001**	0.007	0.019	0.996
MP CT	Correlation coefficient	−0.140	−0.417	0.256	−0.168	0.052
	*p*	0.087	**<0.001**	**0.001**	**0.039**	0.529
MP CI	Correlation coefficient	−0.140	−0.461	0.294	−0.174	0.008
	*p*	0.087	**<0.001**	**<0.001**	**0.032**	0.925
MP CN	Correlation coefficient	−0.105	−0.330	0.167	−0.157	0.006
	*p*	0.201	**<0.001**	**0.040**	0.054	0.942
MP C global	Correlation coefficient	−0.143	−0.434	0.251	−0.226	0.043
	*p*	0.079	**<0.001**	**0.002**	0.005	0.599
Macular Integrity Index	Correlation coefficient	0.180	0.372	−0.220	0.111	0.013
	*p*	0.027	**<0.001**	**0.007**	0.175	0.870
Average total Threshold	Correlation coefficient	−0.212	−0.439	0.249	−0.159	−0.009
	*p*	**0.009**	**<0.001**	**0.002**	0.051	0.913

BCVA: best-corrected visual acuity; SE: spherical equivalent; AL: axial length; IOP: intraocular pressure; MP: microperimetry; SO: superior outer; TO: temporal outer; IO: inferior outer; NO: nasal outer; SI: superior inner; TI: temporal inner; II: inferior inner; NI: nasal inner; C: central point; CS: superior central; CT: central temporal; CI: central inferior; CN: central nasal; C global: central global.

**Table 4 diagnostics-11-01157-t004:** Comparison for MAIA and clinical parameters between macula-on and -off subgroups. Significant differences (*p* < 0.05) are highlighted in bold print.

	Macula-On RRD	Macula-Off RRD	*p*
	Mean +/− SD Median (Range)	Mean +/− SD Median (Range)	
Age (years)	58.95 +/− 11.13 58.95 (33–71)	57.63 +/− 33 57.63 (44–72)	0.188
BCVA (logMAR)	0.13 +/− 0.18 0.126 (0.0–0.6)	0.19 +/− 0.0 0.195 (0.0–0.6)	0.109
SE (diopters)	−2.41 +/− 3.47 −2.41 (−14.00–1.25)	−5.14 +/− −14.00 −5.15 (−13.50–1.50)	0.313
AL (mm)	26.84 +/− 4.55 26.84 (23.68–44.20)	25.39 +/− 23.68 25.39 (21.61–30.79)	0.293
IOP (mmHg)	13.84 +/− 2.27 13.84 (9–17)	15.00 +/− 9 15 (10–18)	0.165
Days between surgery and study protocol	472.95 +/− 449.9 472.95 (19–1458)	435.58 +/− 398.22 435.58 (26–1477)	0.965
Macular Integrity Index	74.47 +/− 29.34 74.47 (6.2–100.0)	86.43 +/− 17.11 86.43 (54.8–100.0)	0.312
Average Total Threshold (dB)	25.56 +/− 2,.78 25.56 (19.0–29.6)	22.99 +/− 5.23 22.99 (4.5–27.8)	**0.041**
Fixation Stability P1 (%)	81.53 +/− 22.96 81.53 (21–100)	77.16 +/− 30.85 77.16 (0–100)	0.930
Fixation Stability P2 (%)	94.11 +/− 10.50 94.11 (61–100)	88.00 +/− 25.34 88 (1–100)	0.761
BCEA 63% area (degrees)	2.45 +/− 3.17 2.45 (0.2–11.3)	6.58 +/− 14.70 6.58 (0.1–62.3)	0.748
BCEA 95% area (degrees)	7.32 +/− 9.51 7.32 (0.5–33.8)	19.73 +/− 44.07 19.73 (0.4–186.7)	0.759
BCEA 63% angle (degrees)	−1.35 +/− 58.95 −1.35 (−86.7–84.6)	−11.37 +/− 48.15 −11.37 (−81.4–81.2)	0.589
BCEA 95% angle (degrees)	−1.35 +/− 58.95 −1.35 (−86.7–84.6)	−11.37 +/− 48.15 −11.37 (−81,4–81.2)	0.589
Fixation losses (%)	3.16 +/− 10.03 3.16 (0–40)	4.47 +/− 11.17 4.47 (0–40)	0.794

RRD: rhegmatogenous retinal detachment; SD: standard deviation; BCVA: best-corrected visual acuity; SE: spherical equivalent; AL: axial length; IOP: intraocular pressure; C global: central global.

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
