# Peer review of "Microperimetry-Assessed Functional Alterations and OCT-Changes in Patients after Retinal Detachment Surgery Using Pars Plana Vitrectomy and SF6 Tamponade"

_diagnostics, 2021, doi:10.3390/diagnostics11071157_

Round 1

Reviewer 1 Report

This is a well-written manuscript. There are only minor concerns:

Abbreviation GCL is not explained in the abstract.

In the title it should be mentioned that results of PPV with gas are shown, as there are possible another surgical techniques.

Author Response

Thank you.

Reviewer 2 Report

the paper is interesting, there are several concerns about previous studies in literature, several studies were performed on Retinal Detachment and microperimetry, in present sudy seems tht the procedure leads a damage in retinal function, in my opinion educed level of retinal sensitivity may be due to detacment,reattachment procedures on the retina. the authors need to be foclized on this asect nor on the BCVA alone that represent only central retinal activity. they must discuss more about fixation and about revoery during the tme alone and with biofeedback stimulation.

Author Response

AUTHOR RESPONSE TO REVIEWERS’ COMMENTS

Dear editor,

We sincerely appreciate your comments and interest in our manuscript entitled "Microperimetry-assessed functional alterations and OCT-changes in patients after retinal detachment surgery using pars plana vitrectomy and SF6 tamponade". We also want to thank you for giving us the chance of trying to solve the reviewers´ comments. We revised the manuscript according to your suggestion and the last change included in the manuscript is detailed below. We are really pleased with our manuscript improvement with all the reviewers´ suggestions and we hope that it would reach the standards of your journal.

We will try to answer point-by-point the editorial comments and the reviewers’ comments. We have also highlighted in red color the changes made in the manuscript.

Best regards,

Isabel Pinilla, MD PhD

Response to the comments of the Reviewer #2:

First of all, we want to thank your comments about the manuscript.

We have added more background to the introduction.

We increased the description of the methods and we changed the Figure 1 trying to improve them.

We have changed the drafting to make the results clearer and easier to understand.

the paper is interesting, there are several concerns about previous studies in literature, several studies were performed on Retinal Detachment and microperimetry, in present sudy seems tht the procedure leads a damage in retinal function, in my opinion educed level of retinal sensitivity may be due to detacment,reattachment procedures on the retina. the authors need to be foclized on this asect nor on the BCVA alone that represent only central retinal activity. they must discuss more about fixation and about revoery during the tme alone and with biofeedback stimulation.

We included in the introduction some details about other functional tests.

We added in the discussion some studies about the biofeedback and changes in PRL that could justify the sensitivity changes. We also added the lack of follow-up as a limitation.

Please, do not hesitate to ask us if you need further information.

Looking forward to your response,

Yours sincerely,

Isabel Pinilla Lozano
